# NF-κB Signaling Regulates Physiological and Pathological Chondrogenesis

**DOI:** 10.3390/ijms20246275

**Published:** 2019-12-12

**Authors:** Eijiro Jimi, Fei Huang, Chihiro Nakatomi

**Affiliations:** 1Oral Health/Brain Health/Total Health Research Center, Faculty of Dental Science, Kyushu University, 3-1-1 Maidashi, Higashi-ku, Fukuoka 812-8582, Japan; 2Laboratory of Molecular and Cellular Biochemistry, Faculty of Dental Science, Kyushu University, 3-1-1 Maidashi, Higashi-ku, Fukuoka 812-8582, Japan; huangfeib@163.com; 3Division of Molecular Signaling and Biochemistry, Department of Health Improvement, Kyushu Dental University, 2-6-1 Manazuru, Kokurakita-ku, Kitakyushu 803-8580, Japan; chihiro19850523@gmail.com

**Keywords:** chondrogenesis, NF-κB, inflammation

## Abstract

The nuclear factor-κB (NF-κB) is a transcription factor that regulates the expression of genes that control cell proliferation and apoptosis, as well as genes that respond to inflammation and immune responses. There are two means of NF-κB activation: the classical pathway, which involves the degradation of the inhibitor of κBα (IκBα), and the alternative pathway, which involves the NF-κB-inducing kinase (NIK, also known as MAP3K14). The mouse growth plate consists of the resting zone, proliferative zone, prehypertrophic zone, and hypertrophic zone. The p65 (RelA), which plays a central role in the classical pathway, is expressed throughout the cartilage layer, from the resting zone to the hypertrophic zone. Inhibiting the classical NF-κB signaling pathway blocks growth hormone (GH) or insulin-like growth factor (IGF-1) signaling, suppresses cell proliferation, and suppresses bone morphogenetic protein 2 (BMP2) expression, thereby promoting apoptosis. Since the production of autoantibodies and inflammatory cytokines, such as tumor necrosis factor-α (TNFα), interleukin (IL)-1β, IL-6, and IL-17, are regulated by the classical pathways and are increased in rheumatoid arthritis (RA), NF-κB inhibitors are used to suppress inflammation and joint destruction in RA models. In osteoarthritis (OA) models, the strength of NF-κB-activation is found to regulate the facilitation or suppression of OA. On the other hand, RelB is involved in the alternative pathway, and is expressed in the periarticular zone during the embryonic period of development. The alternative pathway is involved in the generation of chondrocytes in the proliferative zone during physiological conditions, and in the development of RA and OA during pathological conditions. Thus, NF-κB is an important molecule that controls normal development and the pathological destruction of cartilage.

## 1. Introduction

The mechanism of bone formation is classified into endochondral ossification and intramembranous ossification, which is involved in the formation of flat bones like the skull, and is the mechanism by which the long bones increase in thickness [1,2]. However, many bones are formed by “endochondral ossification”, a process in which cartilage is formed first and then is replaced with bone, and this mechanism is particularly important for bone development. In the fetus, endochondral ossification begins with the condensation of mesenchymal cells that differentiate into chondrocytes to form a type II collagen-rich cartilage template [1,2]. Chondrocytes continue to proliferate and differentiate, forming a growth plate critical for vertical growth. Within the growth plate, proliferating chondrocytes organize into columnar structures, exit the cell cycle, and become hypertrophic chondrocytes. The enlarged cartilage area is calcified and absorbed, and then replaced by the bone and vascular system [1,2].

Endochondral ossification is composed of multiple steps, including condensation of mesenchymal cells; differentiation of these cells into chondrocytes; chondrocyte proliferation, hypertrophy, and apoptosis; subsequent vascular invasion; and calcification, and these complex programs are continuously and exquisitely controlled during development [1,2]. Several proteins, including the bone morphogenetic protein (BMP), the transforming growth factor β (TGF-β), the fibroblast growth factor (FGF), the Indian hedgehog (Ihh), the Wnt family of proteins, the insulin-like growth factor (IGF) and other growth factors, and the parathyroid hormone-related protein (PTHrP), gracefully direct endochondral ossification through the expression of downstream signaling pathway members and chondrogenic transcription factors [1,2]. Notably, some of these transcription factors play important and specific roles in endochondral ossification by regulating the expression of chondrogenic genes. For example, the SRY-box 9 (Sox9) gene is indispensable in the early stages of cartilage formation, but the runt-related transcription factor 2 (Runx2) is required in the later stages [2,3,4].

The transcription factor nuclear factor-κB (NF-κB) regulates the expression of a wide variety of genes involved in immune and inflammatory responses, cell proliferation, tumorigenesis, cell survival, and development [5,6]. Several studies have reported that the NF-κB transcription family is involved in endochondral ossification and limb outgrowth under physiological conditions [7,8,9,10]. 

Inflammation is a host defense response against bacterial and viral infections and tissue damage by alerting, recruiting, and activating immune cells, such as T cells, B cells, and macrophages. [11,12]. NF-κB serves as the transcription factor regulating gene expressions, such as cytokines, chemokines, adhesion molecules, cell cycle regulators, and anti-apoptotic factors in immune cells. Furthermore, NF-κB regulates the expression of pro-inflammatory cytokines and chemokines in macrophages, M1 macrophage polarization, maturation of dendritic cells, and the recruitment of neutrophils. NF-κB also induces the differentiation and activation of T cells and B cells. Thus, inflammation is normally beneficial to the host. However, excessive immune and inflammatory responses induce tissue destruction, including cartilage and bone [11,12]. Rheumatoid arthritis (RA) and osteoarthritis (OA) are the most common human inflammatory diseases accompanied by cartilage destruction [13]. Although the pathological features are different between RA and OA, the pro-inflammatory cytokines, such as interleukin (IL)-1β and tumor necrosis factor (TNF)-α, matrix metalloproteinases (MMPs), and prostaglandins, are involved in the development of both RA and OA [13]. It has been reported that NF-κB is involved in the regulation of these inflammatory activities, and that NF-κB has been implicated in both cartilage differentiation and cartilage destruction. This review describes the physiological and pathological roles of NF-κB in cartilage metabolism.

## 2. The NF-κB Family and Its Signaling

NF-κB was discovered in 1986 as a transcription factor that specifically binds to the enhancer region of the immunoglobulin κ light chain gene [5,6]. It was originally considered a regulator of B cell differentiation and function, but was later found to be a ubiquitous transcription factor in various cells. The NF-κB family of transcription factors is composed of five proteins ubiquitously expressed in mammals: p65 (RelA), c-Rel, RelB, NF-κB1 (p105/p50), and NF-κB2 (p100/p52), which form homodimers and various heterodimers. All five members share a 300-amino acid N-terminal domain called the Rel homology domain (RHD), which is derived from the retroviral oncoprotein v-Rel, which is involved in DNA binding and dimerization and is associated with inhibitor of κB (IκB) proteins. Three members (p65, c-Rel, and RelB) contain C-terminal transcriptional activation domains (TADs) that are important for their ability to induce target gene expression. p65, c-Rel, and RelB are synthesized as mature proteins, whereas NF-κB1 and NF-κB2 are synthesized as large precursor proteins (p105 and p100) that produce mature NF-κB subunits p50 and p52, respectively. The ankyrin repeats in the C-terminal region of p105 and p100 are involved in their degradation along the ubiquitin-proteasome pathway. As a consequence, p50 and p52 homodimers lack TADs and have no inherent ability to drive transcription or act as a transcriptional repressor. However, the heterodimers of p65:p50, c-Rel:p50, and RelB:p52 function as transcriptional activators (Figure 1) [5,6].

IκB proteins tightly regulate the activation of NF-κB signaling. IκB proteins consist of IκBα, IκBβ, IκBε, etc., and the precursors p105 (NF-κB1) and p100 (NF-κB2), which are characterized by multiple ankyrin repeat domains and the ability to bind as NF-κB dimers. In unstimulated cells, NF-κB dimers form a complex with IκB proteins that is maintained in the cytosol. Upon stimulation, IκBs are phosphorylated by IκB kinase (IKK) at conserved serine residues (DSGXXS) and then are degraded by the proteasome. Then, the free NF-κB is translocated into the nucleus where it binds to the promoter region of the target genes (Figure 1) [5,6].

IKK is an enzyme complex that is activated by several stimuli and phosphorylates IκB proteins to modulate the cellular response. The IKK complex consists of three catalytic subunits: IKKα (also known as IKK1); IKK β (also known as IKK2); and a regulatory subunit, NF-κB essential modulator (NEMO), also known as IKKγ (Figure 1). IKKα and IKKβ are structurally similar, but biological and genetic studies have indicated that IKKβ is the main kinase involved in IκB phosphorylation. IKKβ-deficient mice acquired a phenotype similar to that of p65-deficient mice that died by E13.5 from severe liver damage due to the extreme number of cells undergoing apoptosis, indicating the importance of IKKβ in IκB phosphorylation. On the other hand, IKKα-deficient mice die perinatally, with multiple morphological defects. IKKα is mainly involved in the alternative NF-κB pathway. This leads to the processing of p100 to p52, and forms a complex with RelB. Mice lacking NEMO die between E12.5 and E13.0 from severe liver damage due to an extreme number of cells undergoing apoptosis, suggesting that NEMO is indispensable for the activation of NF-κB signaling [5,6].

Activation of NF-κB is regulated by two distinct pathways, referred to as the “classical” and “alternative” NF-κB activation pathways. The classical NF-κB pathway is regulated by IKKβ, which is activated by inflammatory cytokines, such as TNFα, IL-1β, or LPS. In response to a variety of stimuli, IκBα is phosphorylated at serine 32 and 36 by the activated IKK complex, mainly primarily by the IKKβ subunit, and then degraded by the 26S proteasome. In contrast, the alternative NF-κB pathway depends on IKKα and is independent of NEMO. The alternative NF-κB pathway is activated by a select group of TNFR superfamily members, including CD40, the lymphotoxin-β receptor (LTβR), and the receptor activator of NF-κB (RANK). NF-κB-inducing kinase (NIK, also known as MAP3K14) and IKKα are necessary for the processing of p52 from p100, and result in dimerization and activation of the p52/RelB heterodimer (Figure 2) [5,6]. Since these two pathways play different roles, the p50/p65, p50/c-Rel, and the p52/RelB heterodimers are expected to bind to the different promoter region of the target genes. However, the sequence to which p52/RelB specifically binds has not been identified [14].

## 3. The Role of the Classical NF-κB Pathway in Chondrogenic Development

The cartilage structure of the growth plate is composed of four zones: the resting, proliferative, prehypertrophic, and hypertrophic zones [1,2]. Each zone contains chondrocytes at different stages of differentiation. For example, p65 expression is observed throughout the growth plate, but mainly in the resting and hypertrophic zones, suggesting that the classical NF-κB pathway is involved in normal bone development, particularly endochondral ossification [7,10]. Overexpression of a dominant negative form of IκBα that blocks NF-κB activation leads to abnormal limb bud development and downregulation of BMP signaling in chicken embryos [15].

IGF-1-induced NF-κB activation regulates chondrogenesis in growth plates by stimulating chondrocyte proliferation and maturation, and by inhibiting apoptosis [16,17]. Inhibition of NF-κB signaling using PDTC, BAY11-7082, or p65 siRNA reduces chondrocyte proliferation and differentiation, and increases apoptosis by suppressing BMP2 expression, resulting in the suppression of cultured rat metatarsal bone cells and the inhibition of the growth of the metatarsal bone line by growth plate chondrocytes [16,17]. In contrast, overexpression of p65 in cultured chondrocytes induced chondrocyte proliferation and differentiation, and inhibited apoptosis by increasing BMP2 expression. Furthermore, there are two putative NF-κB response elements in the promoter region of the BMP2 gene [18]. These elements were functional in chondrocytes, and NF-κB induced BMP2 expression through these elements. Consistent with these results, the expression of BMP2 in growth plate chondrocytes was dramatically reduced, resulting in a significant decrease in chondrocyte proliferation in NF-κB1 and NF-κB2-double-knockout mice. In postnatal growth plate cartilage, NF-κB-regulated BMP2 plays an important role in cartilage formation [19,20]. Furthermore, recent findings from studies using the ATDC5 chondrogenic cell line indicate the importance of transient NF-κB activation in the initiation of chondrogenic differentiation. BMP2 induces the transient activation of NF-κB during the first few hours of chondrogenic differentiation, and p65 siRNA suppresses BMP2-induced Sox9 expression in ATDC5 cells [21], indicating that, in the early chondrogenic phase of endochondral ossification, BMP2 induces Sox9 expression via NF-κB-activation.

Growth hormone (GH), as well as IGF, stimulates growth plate chondrogenesis and longitudinal bone growth directly at the growth plate [22]. GH was administered to mice lacking the chondrocyte-specific IGF-1 receptor, and the length of the tibia and the growth plate after four weeks of GH treatment was larger than their length in the mice that did not receive GH treatment. GH administration further increased BMP2, NF-κB, and p65 mRNA expression and protein phosphorylation in the growth plate. In addition, some of the effects of GH were mediated by p65. Two different mutations that impair NF-κB activation were reported in two children who, indeed, presented with growth impairments and GH insensitivity [22].

Recently, chondrocyte-specific p65 (RelA)-deficient mice showed mild skeletal growth impairment, an increase in apoptotic chondrocytes, and a significantly decreased number of chondrocytes (Figure 3) [10]. Taken together, these findings suggest that the classical NF-κB signaling pathway might be involved in chondrocyte maturation downstream of IGF-1 and GH, and in the induction of BMP2 by preventing apoptosis.

## 4. The Role of the Alternative NF-κB Pathway in Endochondral Ossification

Although IKKα-deficient mice presented with dwarfism and skeletal defects, the phenotype was caused by disruption of the epidermal-mesenchymal interaction [23,24]. Phosphorylated NF-κB2 and RelB were found in the nucleus in most of the chondrocytes in the periarticular zone but not in the hypertrophic zone of the mouse growth plate at E18.5, suggesting that the alternative NF-κB pathway was activated mainly in the periarticular zone of the growth plate. The *p100^-/-^* mice, in which the homozygous genes encoding the ankyrin repeats in the C-terminus of NF-κB2 were deleted, and in which the DNA-binding activity of the p52/RelB complex was dramatically activated in various tissues, exhibited dwarfism and shortened long bones [25,26]. A histological analysis of the growth plate revealed an abnormal arrangement of chondrocyte rows and an enlarged, narrow area in the bones of these mice. Consistent with these observations, the expression of hypertrophic chondrocyte markers, type X collagen (ColX), and/or matrix metalloproteinase 13, but not early chondrogenic markers, such as Col II or aggrecan, were suppressed in the *p100^-/-^* mice. The in vivo BrdU trace assay clearly showed low proliferative activity in *p100^-/-^* mouse chondrocytes. These defects were partially rescued when the RelB gene was deleted in the *p100^-/-^* mice. These results indicated that the alternative NF-κB pathway regulates chondrocyte proliferation and differentiation to maintain endochondral ossification (Figure 3) [25].

## 5. NF-κB Signaling Is a Target for Preventing Rheumatoid Arthritis (RA)

Rheumatoid arthritis (RA) is a chronic and persistent inflammatory disease that occurs in the joints of the whole body caused by immune abnormalities, and is characterized by the overgrowth and destruction of the synovium in the affected joints [8,27]. In RA, the synovial tissue is significantly thickened, and is infiltrated by inflammatory cells, such as lymphocytes and macrophages. Various cytokines, such as TNFα, IL-1β, IL-6, and IL-17, are produced by these inflammatory cells [11,27,28]. In addition, autoantibodies are produced from these cells, causing a chronic inflammatory response [27,28]. The classical NF-κB pathway regulates the expression of inflammatory cytokines, and is continuously activated by inflammatory cytokine stimulation. Classical pathway factors are considered to be therapeutic targets for RA [27,28]. Previous studies have shown that the constitutively activated classical NF-κB pathway was observed in the synovial tissue of RA patients [29,30,31]. The activation of the classical NF-κB pathway was also observed in joints of animal models, such as mouse collagen-induced arthritis [32] and rat arthritis induced by pristine or streptococcal cell walls [33,34]. Furthermore, adenoviral transfer of IKKβ into rat articular induced NF-κB activation and synovial inflammation, accompanied with clinical symptoms of arthritis [35]. On the other hand, it has been reported that IKKβ-deficient mice [36] and intra-articular gene transfer of the dominant negative form of IKKβ [35] inhibitors, such as NBD peptide, TAT-IκBα -super repressor, and IKKβ inhibitor, etc., [37,38,39,40,41] targeting the classical NF-κB pathway suppressed bone destruction in arthritis models.

NIK is highly expressed in synovial endothelial cells of RA patients and promotes pathogenic angiogenesis and synovial inflammation by inducing CXCL12 [42,43]. NIK is one of the factors that regulates TH17 cell differentiation, and it is considered to be particularly important in autoimmune diseases [28]. NIK^-/-^ mice have been found to be resistant to antigen-induced arthritis resulting from T cell responses [44,45]. B cells producing autoantibodies also involve RA pathogenesis. B cell-activating factors belonging to the tumor necrosis factor family (BAFF), which regulates the survival, differentiation, and antibody production of B cells, activate the alternative NF-κB pathway [28]. Serum BAFF concentration in RA patients is elevated and correlates with rheumatoid factors [46,47]. Furthermore, the concentration of BAFF in synovial fluid of RA patients is higher than that of serum BAFF, suggesting that local BAFF production is involved in the development of RA [46]. BAFF antagonists also improved the arthritis score of collagen-induced arthritis [48]. These results suggest that alternative pathways are involved in the development of RA.

## 6. The Strength of the Classical NF-κB Signaling Regulates Cartilage Homeostasis and Osteoarthritis (OA) Development

Osteoarthritis (OA) is a degenerative disease in joints involving structural alternations in the articular cartilage, subchondral bone, and synovium, with pain and swelling caused by the imbalance between the repair and destruction of joint tissues such as knees, elbows, and shoulders [49,50]. Multifactorial conditions, such as failing joint biomechanics and immune and inflammatory responses, lead to the progression of OA [49,50]. Inflammatory factors affect chondrocyte development. Chondrocytes in OA show increased expression of hypertrophic markers, such as type X collagen, alkaline phosphatase, and MMP13, indicating that OA-derived chondrocytes are mature and differentiated [13,49,50]. Although activation of the classical NF-κB pathway was observed in both RA and OA synovitis, the activation was higher in RA than OA [51]. The knockdown of either IKKα or IKKβ in OA-derived chondrocyte micromass cultures showed increased GAG content, Col2a1 expression, and decreased calcium deposits [52]. Ablation of IKKα resulted in smaller OA-derived hypertrophyic chondrocytes by suppressing Runx2 expression. The expression of IKKα is related to chondrocyte hypertrophy, and the knockdown of IKKα results in decreased synthesis and activity of ornithine decarboxylase that regulates Runx2 expression and nuclear translocation [52]. The silencing of IKKβ markedly enhances accumulation of glycosaminoglycan, in conjunction with SOX9 induction [52]. These results indicate that IKKα and IKKβ exert differential roles in extracellular matrix remodeling and endochondral ossification.

The expression MMPs and tissue inhibitors of MMPs (TIMPs) regulated by inflammatory cytokines degrade and preserve the extracellular matrix produced by chondrocytes, respectively, and they are involved in the onset and development of OA [49,50]. In normal joints, the proportion of TIMPs is greater than that of MMPs, but the amount of MMPs increases due to mechanical stimulation, substrate degradation products, and age-related glycation end products (AGEs) in OA [30,31]. It has been reported that NF-κB is directly involved in the regulation of MMP expression, but a recent study reported that HIF-2α, which is mediated by NF-κB, is involved in the development of OA [10,53]. HIF-2α was identified as a transcription factor that greatly enhances ColX promoter activity in hypertrophic cartilage and enhances the promoter activity of MMP13 and vascular endothelial growth factor (VEGF), as well as Col10A1. In addition, HIF-2α expression is higher in OA cartilage compared to that of normal cartilage in mice and humans, HIF-2α heterozygous mice showed resistance to OA, and a human SNP analysis revealed that HIF-2α is correlated with OA. The results from a HIF-2α promoter analysis revealed that NF-κB regulates HIF-2α transcriptional activity [10,53]. To support these results, injection of the IKK inhibitor BMS-345541 suppressed OA of knee joints by downregulation of NF-κB/HIF-2α signaling in mouse OA models [54].

As described above, defects during the development of the NF-κB p65 subunit cause growth disorders by inducing the apoptosis of cartilage cells. Post-growth depletion of p65 in chondrocytes promotes apoptosis of chondrocytes, and significantly exacerbates OA. However, mice heterozygous for p65 have suppressed OA development. Although the anti-apoptotic genes were not suppressed in either p65 homozygous (wild-type) or heterozygous mice, the expressions of MMPs, VEGF, and inflammation were induced by HIF-2α in the p65 homozygous mice (Figure 4). Taken together, these studies indicate that NF-κB signaling in chondrocytes controls cartilage homeostasis and OA development [10,53,54].

## 7. Conclusions

There are two NF-κB activation pathways: the classical pathway and the alternative pathway. Although these two pathways are known to be involved in the homeostasis of various cells and tissues, the homeostasis of articular chondrocytes is also maintained through the regulation of the proliferation, differentiation, and maturation of chondrocytes via these two pathways. Although in vitro experiments have suggested that TGF-β/BMP activates the classical NF-κB pathway [7,9,50], ligands that activate these two NF-κB activation pathways in vivo and their target genes, which regulate chondrocyte development, are still unknown. On the other hand, the production of inflammatory cytokines, such as TNFα or IL-1β, is increased in RA and OA, in which the joint is destroyed by chronic inflammation. NF-κB inhibitors might be useful for preventing RA or OA joint destruction. However, classical NF-κB activation in osteoarthritis has been reported to have dual effects by both suppressing and enhancing joint destruction. To date, there are no reports regarding the factors that control the strength of the classical NF-κB activation and the role of NIK in the development of OA. Further comprehensive understanding of the molecular events, including articular chondrocyte differentiation, is necessary for preventing joint destruction.

## Figures and Tables

**Figure 1 ijms-20-06275-f001:**
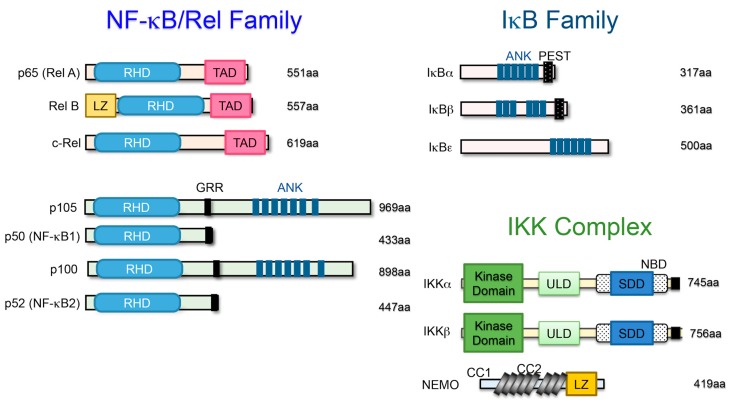
Schematic representation of the nuclear factor-κB/inhibitor of κB (NF-κB/IκB) protein family and the IκB kinase (IKK) family. Members of the NF-κB/IκB protein family and the IKK family are shown. The number of amino acids in each protein is indicated on the right. Presumed sites of cleavage for p105 (amino acid 433) and p100 (amino acid 447) are shown: rel homology domain (RHD), transcriptional activation domain (TAD), leucine zipper (LZ), glycine-rich repeat (GRR), ankyrin repeat (ANK), PEST domain (PEST), coiled-coil domain (CC), ubiquitin-like domain (ULD), scaffolding and dimerization domain (SDD), and NEMO binding domain (NBD).

**Figure 2 ijms-20-06275-f002:**
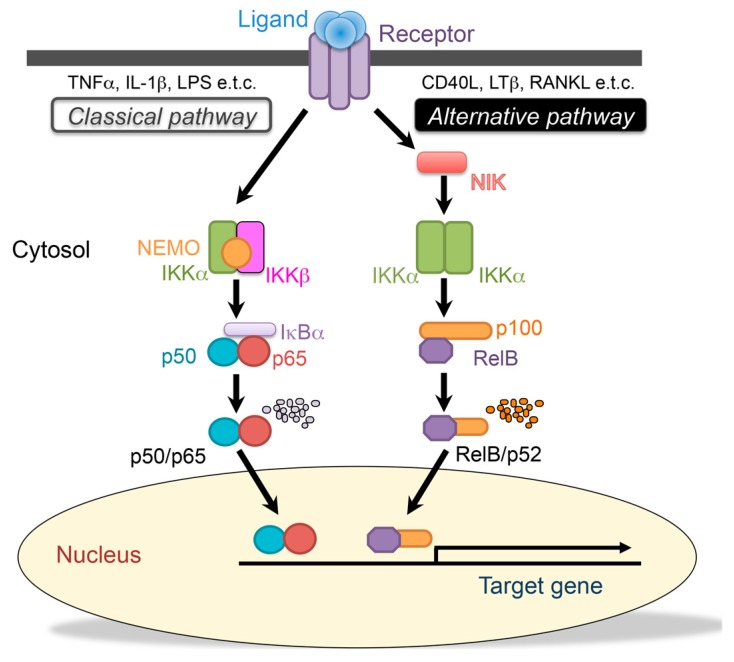
The classical and alternative nuclear factor--κB (NF-κB) signaling pathways. The classical (canonical) pathway is activated by a large number of agonists, such as the tumor necrosis factor-α (TNFα), the interleukin (IL)-1β, lipopolysaccharides, and T-cell receptors. Activation of this pathway depends on the IκB kinase (IKK) complex, which phosphorylates the inhibitor of κBα (IκBα) to induce rapid degradation. This pathway is essential for immune responses, inflammation, tumorigenesis, and cell survival. The alternative (noncanonical) pathway is activated by a limited number of agonists, which are involved in secondary lymphoid organogenesis, mature B-cell function, and adaptive immunity. This pathway requires NF-κB-inducing kinase (NIK) and IκB kinase α (IKKα) to promote the processing of the p100 precursor into p52, which results in dimerization and activation of the p52/RelB heterodimer.

**Figure 3 ijms-20-06275-f003:**
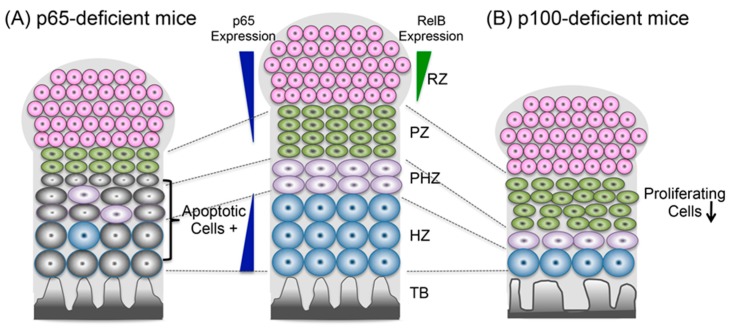
The role of NF-κB signaling in chondrogenic development. (**A**) The classical NF-κB signaling pathway regulates apoptosis (gray cells) in proliferative and prehypertrophic chondrocytes. (**B**) The alternative NF-κB pathway regulates chondrocyte proliferation and differentiation and maintains endochondral ossification. Also shown are the resting zone (RZ), proliferating zone (PZ), prehypertrophic zone (PHZ), hypertrophic zone (HZ), and trabecular bone (TB).

**Figure 4 ijms-20-06275-f004:**
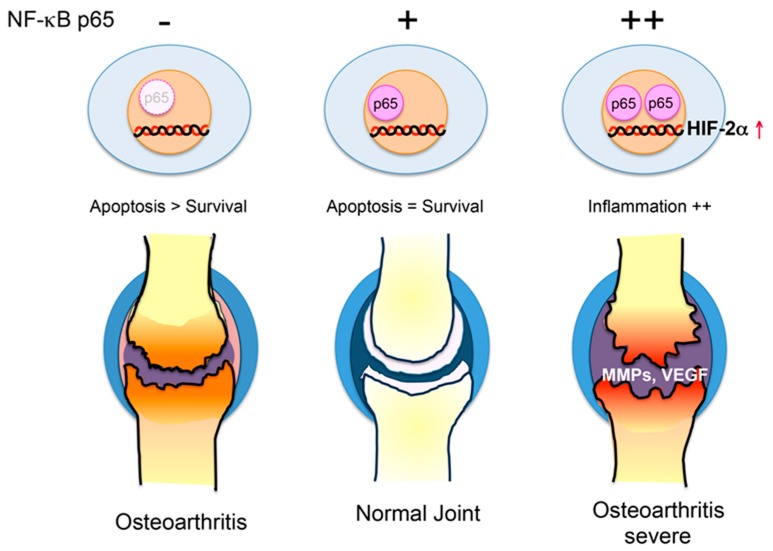
Proposed mechanisms for how NF-κB regulates joint destruction in osteoarthritis. Unregulated NF-κB signaling regulates cartilage homeostasis and osteoarthritis (OA) development. Deficiency of p65 in chondrocytes leads to cartilage degeneration by suppressing anti-apoptotic genes. Moderate NF-κB activity (heterozygous mice) prevents osteoarthritis. Strong NF-κB activity accelerates osteoarthritis by increasing matrix metalloproteinases (MMPs) and the vascular endothelial growth factor (VEGF) via the hypoxia inducible factor 2α (HIF2α).

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
