# Peer review of "NF-κB Signaling Regulates Physiological and Pathological Chondrogenesis"

_ijms, 2019, doi:10.3390/ijms20246275_

Round 1

Reviewer 1 Report

The study by Jimi et al is nice overview of NFKB signaling and its role in endochondral ossification and joint destruction disorders (rheumatoid arthritis, osteoarthritis). In general the work is of high quality. One reservation is that the points for future study indicated at the end of the review are quite generic. Please find more detailed questions and/or comments below.

Is the DNA binding sequence different for the p52/RelB heterodimer when compared to the classical p65:p50, c-Rel:p50 heterodimers?

What is responsible for moderate NFKB activation required for articular cartilage homeostasis? Could the authors speculate about this? Perhaps there is a link with TGF-beta, which is also known to active NFKB signaling. In addition, TGF-beta signaling also differs between normal homeostasis and osteoarthritis. Could this be related to how it activates NFKB?

The authors provide names and common alternative names for most of the crucial components of the NFKB signaling pathway. Does Frontiers encourage the use of HGNC approved nomenclature? For example, the reviewer was unfamiliar with NIK, while the HGNC approved MAP3K14 rings a bell.

In the context of osteoarthritis and hypertrophic differentiation, it might be of relevance to look at a recent review published in Frontiers by Ripmeester et al in 2018.

The final recommendations for future studies are rather a-specific in the opinion of the reviewer: “Further comprehensive understanding of the molecular events, including articular chondrocyte differentiation, is necessary for preventing joint destruction.” Could the authors pinpoint or speculate on what exactly should be further elucidated? For example, the role of NIK in murine osteoarthritis models?

Page 4, line 157; a space should be introduced between p65 and siRNA.

The resolution of Figure 3 is rather low and should be improved.

Page 6, line 236; there seems to be tab introduced after HIF-2a, this should be a space?

Page 7, line 244; NF- ?

Figure 4, perhaps an issue related to lay-out or the reviewers pdf reader, but the figure shows a lot of squares behind every text box. It is not clear what they mean.

Author Response

To reviewer #1,

We greatly appreciate your favorable comments. We are very pleased to know that you have found our review important.

The study by Jimi et al is nice overview of NF-kB signaling and its role in endochondral ossification and joint destruction disorders (rheumatoid arthritis, osteoarthritis). In general the work is of high quality. One reservation is that the points for future study indicated at the end of the review are quite generic. Please find more detailed questions and/or comments below.

Is the DNA binding sequence different for the p52/RelB heterodimer when compared to the classical p65:p50, c-Rel:p50 heterodimers?

Since the classical and the alternative NF-kB activation pathways play different roles, the p50/p65, p50/c-Rel and the p52/RelB heterodimers are expected to bind to their specific DNA sequences. However, the sequence to which p52/RelB specifically binds has not been identified. We added 2 sentences on page4, lines 134-137, and a reference #14 in the revised manuscript.

What is responsible for moderate NF-kB activation required for articular cartilage homeostasis? Could the authors speculate about this? Perhaps there is a link with TGF-beta, which is also known to active NF-kB signaling. In addition, TGF-beta signaling also differs between normal homeostasis and osteoarthritis. Could this be related to how it activates NF-kB?

So far, we have not had the answer for your questions. We added two sentences that we slightly mentioned the relationship TGF-b/BMP signaling with articular cartilage homeostasis on page 9, lines 320-323, and lines 327-329, in the revised manuscript.

The authors provide names and common alternative names for most of the crucial components of the NF-kB signaling pathway. Does Frontiers encourage the use of HGNC approved nomenclature? For example, the reviewer was unfamiliar with NIK, while the HGNC approved MAP3K14 rings a bell.

In the NF-kB filed, all names are in this review are commonly used (Please, see references’ title). In accordance with your suggestion, we added MAP3K14 as another name of NIK on page 1 line 17, and on page 4, lines 132-133, in the revised manuscript.

In the context of osteoarthritis and hypertrophic differentiation, it might be of relevance to look at a recent review published in Frontiers by Ripmeester et al in 2018.

In accordance with your suggestion, we added a reference #50 and cited the reference on page 7 lines 258-265, in the revised manuscript.

The final recommendations for future studies are rather a-specific in the opinion of the reviewer: “Further comprehensive understanding of the molecular events, including articular chondrocyte differentiation, is necessary for preventing joint destruction.” Could the authors pinpoint or speculate on what exactly should be further elucidated? For example, the role of NIK in murine osteoarthritis models?

In accordance with your suggestion, we added 2 sentences on page 9, lines 317-331, in the “Conclusion” section in the revised manuscript.

Page 4, line 157; a space should be introduced between p65 and siRNA.

I apologize our careless mistakes. We added a space between p65 and siRNA on page 5, line 171, in the revised manuscript.

The resolution of Figure 3 is rather low and should be improved.

In accordance with your suggestion, we replaced new Figure 3 in the revised manuscript.

Page 6, line 236; there seems to be tab introduced after HIF-2a, this should be a space?

I could not see any tabs in my computer and printed version. I am not sure what it is, but I deleted the space and then added a space on page 8, line 281, in the revised manuscript.

Page 7, line 244; NF- ?

I apologize our careless mistakes. We corrected “NF-kB” on page 8, line 289, in the revised manuscript.

Figure 4, perhaps an issue related to lay-out or the reviewers pdf reader, but the figure shows a lot of squares behind every text box. It is not clear what they mean.

I could not see any squares in my computer and printed both word and pdf versions. I am not sure what are these, but I converted ppt file to tif file and then attached new Figure 4 in the revised manuscript.

I hope that the revised manuscript would satisfactorily answer the comments raised by you.

Changes made with highlighted have been written in the revised manuscript.

Reviewer 2 Report

Comments

This review aims to provide information on the important roles of transcription factor NF-kB family in both physiological (development, endochondral ossification) and pathological (OA and RA) conditions in chondrocytes. This would be of particular interest to the researchers studying bone metabolism.

The authors describe the gene structures of NF-kB, IkB, IKK, and their two distinct signaling pathways in much detail. However, since ‘signaling’ section and ‘function’ section are not described interrelated with each other, it is hard to understand for readers in what way the differences in the signaling are reflected in the physiological and pathological functions of NF-kB.

Authors should describe the roles of NF-kB in relation to the inflammation, not only to the cell apoptosis and MMPs, because this molecule regulates the expression of many inflammatory genes, and both RA and OA are inflammatory arthritis.

Abstract: Hard to read because the sentences alternate between classical and alternative pathways.

Introduction: The authors should provide more detailed information regarding the involvement of NF-kB in the inflammatory responses and resulting tissue destruction, citing relevant literatures (lines 63-66).

Line 92: Figure 2 instead of Figure 1?

Line 199: Reference?

Figure 3: Authors should define the color of apoptotic cells shown in the diagram in the figure legend.

Line 212-214: ‘infiltrated by various cytokines’ does not make sense. reference?

Line 214-215: The sentence is not grammatically accurate.

Line 216: Reference?

Line 220: Authors should name the inhibitors.

Line 222: Reference?

Line 232-235: The sentence does not make sense.

Line 226-251: Authors should research the literatures from other than this particular laboratory.

Figure 4: ‘Moderate NF-kB activity’ means ‘normal NF-kB’?

Author Response

To reviewer #2,

We would like to thank you for your supportive comments, which we think greatly enhanced our revised manuscript.

This review aims to provide information on the important roles of transcription factor NF-kB family in both physiological (development, endochondral ossification) and pathological (OA and RA) conditions in chondrocytes. This would be of particular interest to the researchers studying bone metabolism.

The authors describe the gene structures of NF-kB, IkB, IKK, and their two distinct signaling pathways in much detail. However, since ‘signaling’ section and ‘function’ section are not described interrelated with each other, it is hard to understand for readers in what way the differences in the signaling are reflected in the physiological and pathological functions of NF-kB.

In accordance with your suggestion, we added 7 sentences in the “rheumatoid arthritis” section, on page 7, lines 232-237, 242-243, 246-253, and a paragraph on page 7, lines 259-275, in the “osteoarthritis” section in the revised manuscript.

Authors should describe the roles of NF-kB in relation to the inflammation, not only to the cell apoptosis and MMPs, because this molecule regulates the expression of many inflammatory genes, and both RA and OA are inflammatory arthritis.

In accordance with your suggestion, we added 7 sentences in the “Introduction” section on page 2, lines 63-75, in the revised manuscript. We also modified Figure 4 in the revised manuscript.

Abstract: Hard to read because the sentences alternate between classical and alternative pathways.

In accordance with your suggestion, we mentioned regarding the role of the classical NF-kB activation pathway and then, followed by the alternative NF-kB activation pathway in the “Abstract” section in the revised manuscript.

4. Introduction: The authors should provide more detailed information regarding the involvement of NF-kB in the inflammatory responses and resulting tissue destruction, citing relevant literatures (lines 63-66).

As described above, in accordance with your suggestion, we added 7 sentences in the “Introduction” section on page 2, lines 63-75, in the revised manuscript. We also modified Figure 4 in the revised manuscript.

5. Line 92: Figure 2 instead of Figure 1?

I apologized our careless mistakes. I corrected “Figure 2” to “Figure 1” on page 3, line 104, in the revised manuscript.

6. Line 199: Reference?

In accordance with your suggestion, I added a reference #26 in the revised manuscript.

Figure 3: Authors should define the color of apoptotic cells shown in the diagram in the figure legend.

In accordance with your suggestion, we modified that apoptotic cells were drawn as gray cells to distinguish living cells (color cells) in the new Figure 3 in the revised manuscript.

Line 212-214: ‘infiltrated by various cytokines’ does not make sense. reference?

In accordance with your suggestion, we rewrote a sentence on page 6, lines 224-226, and added a reference #27 in the revised manuscript.

Line 214-215: The sentence is not grammatically accurate.

In accordance with your suggestion, we rewrote a sentence on page 6, lines 226-228, in the revised manuscript.

Line 216: Reference?

In accordance with your suggestion, we rewrote a sentence on page 6, lines 226-228, and added 3 references #11, 27, 28 in the revised manuscript.

Line 220: Authors should name the inhibitors.

In accordance with your suggestion, we added name of inhibitors on page 7, line 239, in the revised manuscript.

Line 222: Reference?

Is it line 226 in the original manuscript? If so, we added a reference #28 in the revised manuscript.

Line 232-235: The sentence does not make sense.

In accordance with your suggestion, we rewrote a paragraph on page 7, lines 258-274, in the revised manuscript.

Line 226-251: Authors should research the literatures from other than this particular laboratory.

In accordance with your suggestion, we added 4 sentences and 3 references #46, 47, 48 on page 7, lines 246-253, in the revised manuscript.

Figure 4: ‘Moderate NF-kB activity’ means ‘normal NF-kB’?

This is based on the paper from Kobayashi H et al (reference #10). They used p65 homozygous (wild-type), heterozygous, and knockout mice in their experiments. “Moderate NF-kB activity” means “heterozygous”. We drew “-“ (no p65), “+” (a p65) and “++” (2 p65 ) in the Figure 4 in the original manuscript. We modified 2 sentences on page 8, lines 294-298 and added “heterozygous mice” in the “Figure Legends” in Figure 4 in the revised manuscript.

I hope that the revised manuscript would satisfactorily answer the comments raised by you.

Changes made with highlighted have been written in the revised manuscript.

Round 2

Reviewer 2 Report

Line 64; ‘etc.’ instead of ‘e.t.c.’

Line 272; ‘Silencing’ should be ‘silencing’

Line 295; ‘not suppressed in either p65 homozygous (wild-type) or heterozygous mice,’ ?

Author Response

To reviewer #2,

We greatly appreciate your favorable comments and careful review.

I deeply apologize my careless mistakes. I corrected “e.t.c” to etc on page 2, line 64, “Silencing” to “silencing” on page 7, line 271, and “‘not suppressed in either p65 homozygous (wild-type) or heterozygous mice” on page 8, line 294, in the revised manuscript.

I hope that the revised manuscript would satisfactorily answer the comments raised by you.

Changes made with highlighted have been written in the revised manuscript.